# Moral judgment of objectionable online content: Reporting decisions and punishment preferences on social media

Sarah Vahed[1]*, Catalina Goanta[2], Pietro Ortolani[3], Alan G. Sanfey[1,4]

1 Donders Institute for Brain, Cognition and Behaviour, Radboud University, Nijmegen, The Netherlands, 2 Faculty of Law, Utrecht University, Utrecht, The Netherlands, 3 Faculty of Law, Radboud University, Nijmegen, The Netherlands, 4 Behavioural Science Institute, Radboud University, Nijmegen, The Netherlands

* sarah.vahed@donders.ru.nl

**Data Availability Statement:** All measures and stimuli used in this study are described in the respective Method sections and in the Supplemental Material. Open data and materials

## Abstract

Harmful and inappropriate online content is prevalent, necessitating the need to understand how individuals judge and wish to mitigate the spread of negative content on social media. In an online study with a diverse sample of social media users (n = 294), we sought to elucidate factors that influence individuals' evaluation of objectionable online content. Participants were presented with images varying in moral valence, each accompanied by an indicator of intention from an ostensible content poster. Half of the participants were assigned the role of user content moderator, while the remaining participants were instructed to respond as they normally would online. The study aimed to establish whether moral imagery, the intention of a content poster, and the perceived responsibility of social media users, affect judgments of objectionability, operationalized through both decisions to flag content and preferences to seek punishment of other users. Our findings reveal that moral imagery strongly influences users' assessments of what is appropriate online content, with participants almost exclusively choosing to report and punish morally negative images. Poster intention also plays a significant role in user's decisions, with greater objection shown to morally negative content when it has been shared by another user for the purpose of showing support for it. Bestowing a content moderation role affected reporting behaviour but not punishment preferences. We also explore individual user characteristics, finding a negative association between trust in social media platforms and reporting decisions. Conversely, a positive relationship was identified between trait empathy and reporting rates. Collectively, our insights highlight the complexity of social media users' moderation decisions and preferences. The results advance understanding of moral judgments and punishment preferences online, and offer insights for platforms and regulatory bodies aiming to better understand social media users' role in content moderation.

have been made publicly available on OSF at https://osf.io/4ygj3/.

**Funding:** The author(s) received no specific funding for this work.

**Competing interests:** The authors have declared that no competing interests exist.

## Introduction

With billions of daily users across the world, there can be little doubt that social networking sites (SNSs) or platforms, such as YouTube, Facebook and Twitter/X, have become an integral part of society [1]. Despite the benefits of globalised digital communities, the rise in social media use has also resulted in a dramatic increase in incidences of online harm, and the resulting need to moderate such content [1, 2]. Depictions or promotions of psychologically and physically damaging behaviours, attitudes, or experiences encompass the broad spectrum of online content which may be considered harmful [3, 4]. The consequences of exposure to such content are varied, with a large body of research associating exposure to negative online content with, for example, increased self-harm behaviours and suicidal ideation in vulnerable populations [5], elevation of extreme violence within communities [6], and widespread dissemination of fake news, terrorist propaganda and ensuing political polarisation across society [7, 8]. Consequently, the regulation of online harm has risen high on the agenda of social media platforms and has been deemed to one of the greatest concerns for governments, supranational bodies, and international organisations [9, 10].

As a means of combating harm online, SNSs allow users to submit complaints, primarily via reporting mechanisms (also known as 'notice-and-action' mechanisms), that inform the platform of the need to act against content. These actions can include removing content, making it inaccessible, and/or suspending the accounts of those who posted it [9]. Using reporting mechanisms, individual users can express their objection to what they see, and can thus play an important role in the content moderation process. This has been identified as a key means of reducing harm online, as is evidenced by recent legislation adopted in the European Union (EU), specifically the Digital Services Act (DSA), which aims to reform platform governance, including through the standardization of notice-and-action mechanisms across SNSs. Legal rules are also laid out on how posters of harmful content can be penalised by platforms, who hold the unique position of recording and evaluating user behaviours.

Given that social media users are exposed to problematic posts and can play a significant role in identifying harm, it is especially important to understand the factors which influence their decisions to flag online content. Despite this, much of the debate regarding content governance remains focused on the duties and obligations of SNSs, with limited empirical efforts to investigate the responses of individual users. Specifically required is a clear understanding of the content that users think should be removed from platforms and their preferences regarding how other users should be punished. Therefore, this study examines objections to social media content by experimentally investigating factors that drive responses of everyday users to various types of image content on SNSs, advancing knowledge regarding content moderation in novel and useful ways.

### Drivers of objectionability on social media

A key aspect to better understanding how the public responds to and is impacted by social media is investigating what users themselves deem to be objectionable online content. Across SNSs, posts that amount to harassment, hate speech, bullying, as well as depictions of self-harm, suicide or extreme violence are just some examples of content that platforms have considered to be harmful [9]. While some categories may be rendered illegal under national or supranational laws, platforms are often also expected to tackle harmful content that may not be illegal. A common feature of such harmful but lawful content is their moralised nature, that is, they may be considered as morally evocative through their promotion of immoral behaviours or depiction of harmful acts. Importantly, recent studies have found that social media communication containing expressions of morality are actually prioritized by users, resulting

in their associated increased virality [11, 12]. Given this emphasis on moralised content, and the danger potentially posed to individuals and society at large, there is therefore a clear need for better conceptualization of how, and why, people seek to remove certain types of online content. To date however, there is scant empirical evidence as to whether different types of moral content directly impact users' perceptions of objectionability. Moreover, prior research investigating virality of online content has primarily tested user responses to moralized text as opposed to imagery, though of course the latter comprises a large portion of the content which users see on social media [11]. Therefore, in this study we specifically seek to determine whether morality depicted via *images* on social media impact objectionability concerns.

In addition, important contextual factors may impact users' assessments of what is or is not appropriate. For example, a representation of violence can be shared by someone with the intention of promoting the depicted behaviour, but can also be shared by a user with the aim of spreading awareness against, or denouncing, such harmful act. This suggests that the *intentions* of other users (or content posters) may be an important factor in decisions to flag content. Our sensitivity to the intentions of others has, in fact, been described as a core tenet of moral judgment, with prior research highlighting how individuals base their moral judgments on agents' *intentions* and not solely on the outcome of their actions [13–15]. For example, someone who deliberately causes slight harm is typically considered to be more blameworthy than someone who is accidentally responsible for more serious harm [13, 15]. This relevance of intention is also seen in various practical settings, including in the legal assignment of punishment for the infliction of harm [16]. In line with prevailing conceptions of justice, including philosophies of retributive justice, a person who intentionally kills another would (in the absence of a valid legal defence) be found guilty of murder, whereas a person who accidentally causes the death of another may be found guilty and punished for the less serious charge of involuntary manslaughter [17]. Therefore, regardless of the amount of harm done, it is often the intention of an agent, and not only the outcome of their behaviour, that is morally (and legally) relevant. Although a large body of research has focused on explaining how individuals judge and punish what is right or wrong in the physical world [14], how these judgments are shaped in the digital world of social media is still poorly understood. In light of literature emphasising the relevance of intention [14], here we examine the impact of an important contextual factor, namely the intention behind the content itself, on the decisions of individuals to object online.

Whilst it is important to understand *what* users determine to be inappropriate online, it is equally necessary to explore the circumstances under which they decide to mitigate its spread. Over several decades social scientists have attempted to understand related questions about when and why individuals intervene to stop or prevent harm [18]. Empirical studies have consistently identified a bystander's *perceived responsibility* for intervention as a key determinant of their action versus inaction [18, 19]. In this context, perceived responsibility refers to an individual's subjective assessment of their sense of obligation to deal with an incident of harm [19, 20]. In their seminal work on the topic, Darley and Latané suggest a bystander intervention model whereby an individual is less likely to act to stop harm when they assign responsibility to intervene to others (for example, law enforcement or higher authorities) or diffuse such obligation amongst others (for example, anyone else present). This bystander effect is not only one of the most well-established precepts in the field of social psychology, but also one of the most frequently replicated and robust in the assessment of face-to-face contexts [18, 21]. Recent research has now begun to shed some light into how bystander intervention ideals and practices could impact the virtual world by examining responses to witnessing harm online [18, 20]. Drawing upon the bystander effect, research has shown that witnesses of social media harassment who believe they have the responsibility to report incidents are more likely to

adopt coping strategies aimed at resolving the adverse situation and are then more likely to actively help [20]. However, individual responses to online victimization are diverse, and the response and factors that influence them may vary by victimization type and situational characteristics [18]. Obermaier and colleagues [22] found that the more severe a cyberbullying incident was (as described to college student participants in an online experiment), the more likely they were to view the situation as an emergency and, in turn, the more likely respondents said they would be to intervene. Despite initial empirical support however, the degree to which perceived responsibility to intervene influences online bystander intervention behaviour has not been exhaustively examined. This is particularly important given the proliferation of platform affordances which are aimed at community intra-group moderation, such as the possibility of appointing chat moderators, using flagging features to bring certain content to their attention, or even allowing channel/account administrators to create lists of banned terms which would automatically lead to the filtering out of comments inclusive of such terms [23]. Accordingly, here we extend previous work [19, 22], specifically bestowing content moderation responsibility on some participants and examining how this influences views about objectionable content.

Lastly, beyond the content posted on social media, individual users themselves may vary in their perceptions of what constitutes objectionable content online and how they respond to it. This variation in perception and response can be influenced by a range of individual characteristics, with prior research highlighting the role of factors such as trust and empathy in shaping individuals' views on objectionability [24, 25]. Trust can affect how people evaluate others' actions and their willingness to report such behaviours with individuals who trust authorities that handle reports of objectionable behaviour more likely to report such behaviour than those who do not trust these entities [25, 26]. On the other hand, empathy can influence how people emotionally and cognitively respond to objectionable acts and their views on punishment as an appropriate response. For example, highly empathetic people may be more likely to experience emotional distress in response to objectionable acts and support punishment to address harm [24]. Accordingly, we explore the role of trust and empathy in shaping views on objectionability on social media, discussing how these findings contribute to understanding individual differences in such perceptions in online environments.

## Capturing objectionability on social media

In order to effectively measure objectionability online, it is worth noting that the content moderation process relies on important, yet distinct, judgments that aid in curtailing harm online. These judgments include user decisions to report content and platform decisions to moderate that content, for instance by suspending the account of the user that posted it. Similarly, research in moral psychology distinguishes conceptually between types of objectionability, including judgments of wrongness and judgments of punishment [14, 27]. While judgments of wrongness refer to an individual's determination of whether an action or behaviour is morally inappropriate or unacceptable, judgments of punishment serve a regulatory or pedagogical function referring to an assessment of the appropriate consequence or penalty for such wrongful act [14]. The two decisions are thus related, with the former specifically focused on the moral evaluation of an action and the latter focused on identifying a suitable response to that action. Capturing these separate but related measurements in users provides valuable and useful information on the public's perceptions about harmful content and how it should be addressed. Therefore, in this study, we operationalise social media users' objection to online content by two distinct measures considered in both moral psychology and content moderation policies: decisions to report content and preferences to punish content posters.

### The present study

Given the growing need to address online harm, and the limited empirical work that directly investigates factors driving social media users' objectionability to content, this study specifically seeks to answer the following research questions: What type of moralised content do individuals think should be reported and punished on social media? What is the role of intention behind sharing an image on reporting and punishment decisions? Does an individual's sense of responsibility impact decisions to flag content and punish other users?

Based on prior work which has highlighted the prioritisation of morally polarising content [11, 28], our hypothesis is that users focus on moral messaging, particularly those which are morally negative, when making decisions to object to online content. Moreover, given the abundance of research which has emphasised the importance of perceived intent in moral judgments [14, 27], we hypothesise that poster intention will emerge as a significant determinant in assessments of online objectionability. Finally, acknowledging the robust impact of the bystander effect [18, 19], we anticipate that altering the mindset of the user as to their role in content governance will impact their willingness to intervene through moderation mechanisms. We additionally investigate how these user decisions and preferences may be understood by individual characteristics of social media users, such as their trust in SNSs and their empathy.

## Methods

### Participants

The sample size was calculated using $G^*Power$ 3.1 [29], which estimated $n = 119$ to achieve a power of 0.95, with significance level of $p = 0.05$, and a model featuring three test predictors. Considering the absence of prior research on reporting and punishment decisions on social media, we opted for a conservative approach by collecting a sample size of 150 per between-subject group. 300 participants were recruited through Prolific (www.prolific.co), and directed to an online experiment programmed on Gorilla (https://gorilla.sc).

To enrol in the study, participants were required to be 18 years or older, possess (self-reported) fluency in the English language, reside in an EU member state, and be a self-reported active social media user (as defined to mean the use of one or more platform at least once a month). This criteria ensured we obtained a representative sample of social media users across diverse countries in the EU that are subject to content governance legislation, in this case, the DSA. At the end of the experiment, participants were financially compensated for their time through the Prolific platform ($M_{payment}$ = 10.04GBP/hr; $M_{time}$ = 25minutes 5seconds).

Data from 6 participants were excluded, with 2 excluded due to exceeding the assigned time limit of the study, and the remaining 4 as a result of failing more than 25% of the attention checks embedded within the experiment. All of the remaining participants ($n = 294$; 170 males, 123 females and 1 'prefer not to say'; $M_{age}$ = 28.33 years, $SD_{age}$ = 8.42) were included in the analyses (sample size per 'Group' as defined below: Control: $n = 149$; Responsibility: $n = 145$).

### Ethics statement

Prior to data collection, ethical approval for the study was provided by the Ethics Committee Faculty of Social Sciences, Radboud University (**ECSW-2022-011**), with data collection taking place on 7 and 8 June 2022. Prior to the start of the experiment, participants were informed of the general, as well as the sensitive, nature of the study and provide their express, written consent. All measures and stimuli used in this study are described in the respective Method

sections below and in the Supporting Information. Open data and materials have been made publicly available on OSF at https://osf.io/4ygj3/.

## Materials

A total of 66 moral images were obtained from the Social-Moral Images Database (SMID), which contains images representing a wide range of morally positive, negative, and neutral content [30]. For each image, normative ratings are available, including in respect of the level of 'Moral Wrongness'. We selected 22 morally negative, 22 morally positive and 22 morally neutral images. In our selection of "morally negative" images, we specifically excluded images which were extremely graphic, sexually explicit and/or shocking (for example, naked or mutilated bodies). The morally negative, positive and neutral images are hereafter collectively referred to as the '**Images**'. Moral Wrongfulness scores for all images and selection procedure are presented in the Supporting Information.

Each of the Images was accompanied by an indication of intention, namely that the person who chose to share the image (the poster) either approved or disapproved of it. To indicate poster approval, intention was depicted by both a thumbs-up icon and the following sentence: "*The poster liked this image*". To reflect poster disapproval, intention was shown as a thumbs-down icon and the following sentence: "*The poster disliked this image*". The combined icon and sentence are hereafter collectively referred to as the '**Poster Intention**'. In line with methods employed in previous literature [31], two versions of the social media content were created: In Version 1, half of the photos in each category were displayed with poster approval and half were displayed with poster disapproval. In Version 2, the displayed intention was the opposite as that in Version 1 (i.e. if an image was displayed with approval in Version 1, it was displayed with disapproval in Version 2). Half of the participants within each group (as described below) saw Version 1 and the other half saw Version 2; this allowed us to hold the Images constant while manipulating Poster Intention. Fig 1(A)–1(C) shows representative content used in this study.

Prior to commencing the experimental tasks, participants were randomly assigned into one of two groups. Participants in the 'Control' group were instructed that they would rate real social media posts and should respond using the criteria they would personally employ when responding to content they see on social media. These participants were specifically informed that there were no right or wrong answers to the tasks, and were thus not endowed with any specific responsibility to moderate content. Participants in the 'Responsibility' group were similarly instructed that they would be rating real social media posts, but here were assigned the role of 'user content moderator'. These participants were specifically informed that they had the responsibility of identifying inappropriate online content throughout the experimental tasks. Beyond this instruction, no other incentive to moderate content was provided to these participants. The 'Control' and 'Responsibility' groups are hereafter collectively referred to as the '**Groups**'.

Participants' social media usage were captured by recording how often they use 12 popular social media platforms on a 6-point Likert Scale ranging from '*Multiple times a day*' to '*Never*'. Trust in each social media platform was captured by participants reporting how much they agree with the statement "I trust this social media platform" in respect of each of the 12 platforms, paired with a 5-point Likert Scale ranging from '*Strongly Agree*' to '*Strongly Disagree*'. Participants' prior exposure to negative content and self-reported preferences to punish others for posting harmful content was captured using a 5-point Likert Scale response ranging from '*Strongly Agree*' to '*Strongly Disagree*'. Participants' (self-reported) political orientation was measured on 7-point Likert Scale ranging from '*Very Liberal*' to '*Very Conservative*'. Moreover,

## Example Stimuli

## Experimental Tasks

**Fig 1. Stimuli and tasks.** a—c: Representative 'social media' content (representative images taken from the Socio-Moral Image Database (SMID), [30]): (a) Morally positive image with poster approval; (b) Morally neutral image with poster disapproval; and (c) Morally negative image with poster approval. d & e: Judgment and Punishment tasks: (d) Judgment task with choice options ('Like', 'Dislike' or 'Report'); (e) Punishment task with slider ranging from zero-day punishment (No Ban) to 30+ day punishment (Permanent Ban).

three subscales from the Interpersonal Reactivity Index (IRI) were used for the multi-dimensional assessment of empathy, specifically 'Empathic Concern', 'Perspective Taking', and 'Personal Distress'. Items in each subscale were paired with a 5-point Likert response scale (0 = "*Does not describe me well*" to 4 = "*Describes me well*") [32]. IRI's fourth subscale of 'Fantasy', which assesses the ability to imagine and experience the emotions of fictitious characters, was not relevant to our topic of interest and was not included. See Supporting Information for questionnaires included in the study.

## Procedure

Participants completed two tasks–a Judgment task and Punishment task–followed by post-task questionnaires and debriefing. To begin, each participant saw 66 trials consisting of an Image and Poster Intention presented in random order. On each trial (see *Fig 1D*). Participants

were instructed to choose one of three options: '*Like*', '*Dislike*' or '*Report*'. Participant were instructed that the '*Like*' option should be used to reflect content that the participant likes and thinks is okay to be shared online. The '*Dislike*' option should be used to reflect content that the participant dislikes but thinks is acceptable to be shared online. Finally, the '*Report*' option should be used to reflect content that the participant thinks should be removed from the internet. After each trial, a fixation cross was presented for 100 ms, with a self-paced break halfway through. To ensure comprehension of instructions and attentiveness, four practice trials were conducted at the start and two attention checks were presented during the course of the task.

Following the Judgment task, participants were instructed that they would view posts which had previously been reported during the Judgment task either by them or by other participants of the study, and that they would have the opportunity to assign punishment to the poster of that content. In the 'Responsibility' group they were specifically reminded that they would be completing this in their user content moderator role. Each participant saw 36 trials (which were preselected from the 66 trials of the Judgment task), presented in random order (see Fig 1E). On each trial, participants were instructed to specify how long they would like to ban the content poster for, by moving a slider between '0 days (No ban)' and '30+ days (Permanent ban)'. The starting point of the slider was randomized on each trial. After each trial, a fixation cross was presented for 100 ms, with a self-paced break halfway through. To ensure participant comprehension and attention, four practice trials were conducted at the start and two attention checks were presented during the course of the task.

After completing the two tasks, participants answered questions concerning their social media behaviour and opinions as well as their empathy traits in fixed order.

## Statistical analyses

All statistical analyses were performed in *R* (R Core Team, version 3.4.4., 2018). In instances where assumptions for parametric tests were violated, non-parametric equivalents were employed.

In respect of the Judgment Task, we fitted a multilevel logistic regression to analyse the decision to report (binary: *Report versus Not Report* (i.e. '*Like*' and '*Dislike*' choices combined)). We used the *glmer* function from the *lme4* package in R with a binomial distribution and a logit link function [33] (R Core Team, 2017). The fixed effects in the model included three categorical predictors: Image, Poster Intention, and Group, which were treated as factors. A random intercept for each participant (Participant) was included to account for the repeated measures design and the dependency of observations within each participant [34]. The model formula was specified as follows: Report ~ factor(Image) + factor(Poster Intention) + factor (Group) + (1 | Participant).

Furthermore, we modelled the data in the Punishment Task using a generalized linear mixed model with a gamma distribution and a log link function, using the *glmmTMB* package in R (R Core Team, 2017). The distribution choice was made due to its ability to capture the skewed nature of the dataset. We first transformed the dependent variable (punishment amount (continuous: 0–30)) by adding a small constant (0.001) to all values to ensure they were strictly positive, as required for the gamma distribution. The fixed effects in the model included three categorical predictors: Image, Poster Intention, and Group, which were treated as factors. A random intercept for each participant (Participant) was included to account for the repeated measures design and the dependency of observations within each participant. The model formula was specified as follows: Response ~ factor(Image) + factor(Poster Intention) + factor(Group) + (1 | Participant).

## Results

### Judgment task

Across all participants, 59.89% of the content was liked, 25.68% was disliked and 14.45% was reported. Notably, participants in the Responsibility group reported more content (7.7%) than those in the Control group (6.7%). In respect of the image categories, the majority of responses to positive images was "Like" (91.65%) with fewer choices to "Dislike" (8.1%) and "Report" (0.26%). Similarly, the majority of responses to neutral images was "Like" (79.6%) with fewer decisions to "Dislike" (20.18%) and "Report" (0.25%). In contrast, 48.76% of negative images were disliked and 42.83% were reported, with only 8.41% liked. Fig 2A displays mean and

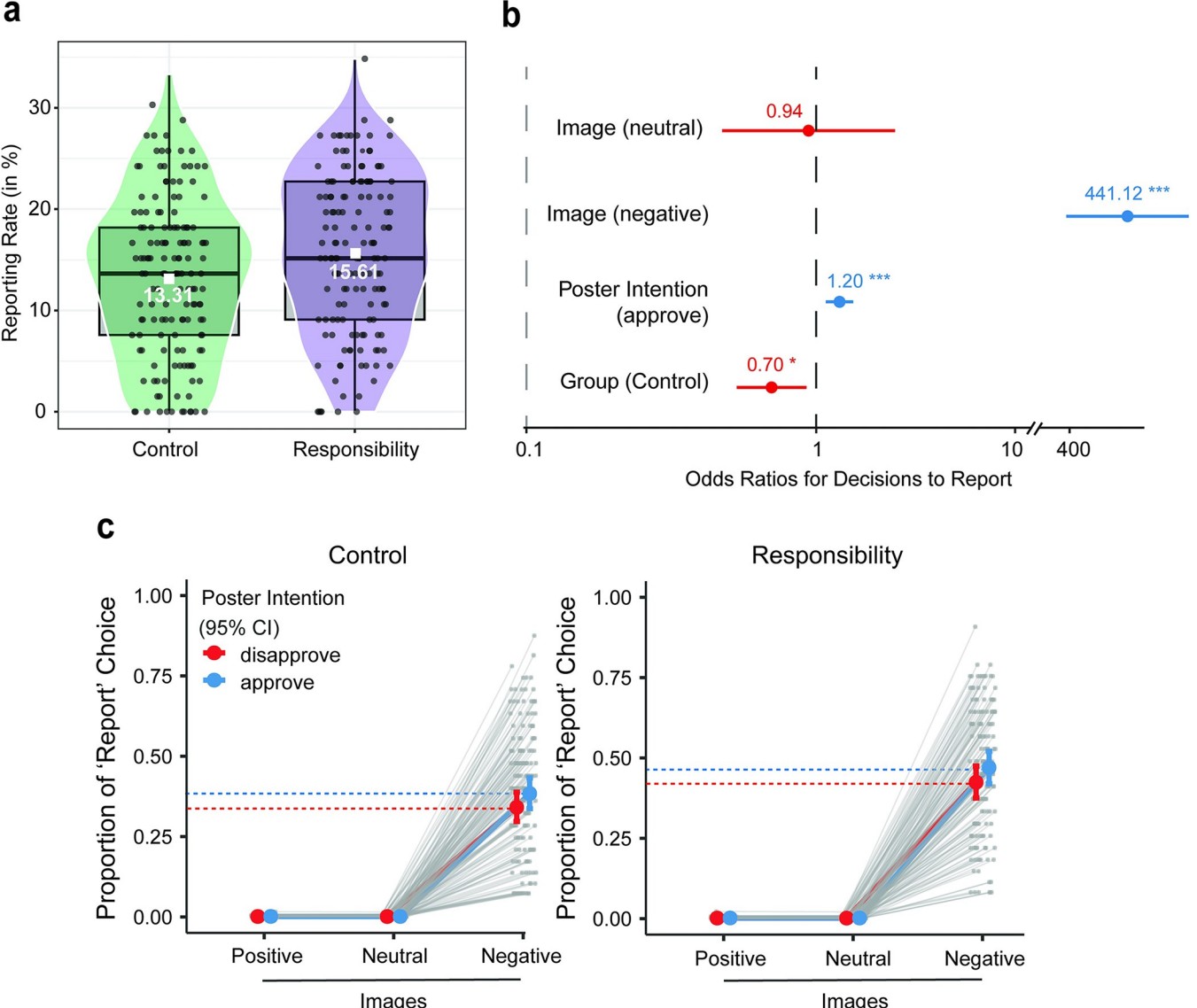

**Fig 2. Judgment task behaviour.** (a) Median reporting rates between Groups, visualised using boxplot and violin plot with jittered data points. The y-axis represents reporting rate (in %), and the x-axis shows the two Groups. The white box indicates mean values and violin plot displays the distribution of data. (b) Model estimates expressed as odds ratio for each of the model parameters. Significance ***p < .001; *p < .05. (c) Proportion of decisions to 'Report' across Images and Poster Intention, between Groups. Error bars represent 95% Confidence Intervals.

**Table 1. Logistic regression model output for the likelihood of reporting social media content, showing fixed effects estimates, standard errors, z-values, and p-values.**

| Independent variable | $\beta$ | SE | z-value | p-value |
|---|---|---|---|---|
| Image: neutral | -0.061 | 0.350 | -0.173 | 0.86 |
| Image: negative | 6.089 | 0.248 | 24.578 | < .001 *** |
| Poster Intention: approve | 0.185 | 0.055 | 3.358 | < .001 *** |
| Group: Control | -0.354 | 0.141 | -2.509 | 0.01 * |

median reporting rates (captured through percentage of all content reported) between Groups.

Logistic regression analysis revealed that the likelihood of choosing to Report social media content was higher for 1) Image category, specifically morally negative compared to morally positive images ($OR$ = 441.11; 95% CI [271.07,717.81]; $p <$ .001), but not for morally neutral compared to morally positive images ($OR$ = 0.94; 95% CI [0.47,1.87]; $p$ = 0.86)); 2) for Poster Intention, specifically poster approval compared to poster disapproval ($OR$ = 1.20; 95% CI [1.08,1.34]; $p <$ .001); and for 3) 'Responsibility' compared to 'Control' groups ($OR$ = 0.70; 95% CI [0.53,0.93]; $p$ = 0.01) (See Table 1). Fig 2B shows model estimates expressed as odds ratios for each of the model parameters, and Fig 2C shows the likelihood of decisions to report content across Images and Poster Intention, and between Groups. Further analysis was conducted to test the association between decisions to Report and Poster Intention for the subset of negative images. Of the negative images that were reported, 47.7% had been shared with poster disapproval, while 52.31% had been shared with poster approval. A chi-square test of independence was performed which revealed a statistically significant association between decisions to Report and Poster Intention, specifically in the context of negative images ($\chi^2(1)$ = 5.91, $p$ = 0.015).

## Punishment task

Participants chose to punish content posters of negative images ($M_{punish-negative}$ = 11.53 days, $SD_{punish-negative}$ = 12.18) more than those who had posted positive ($M_{punish-positive}$ = 0.37 days, $SD_{punish-positive}$ = 2.37) or neutral images ($M_{punish-neutral}$ = 0.41 days, $SD_{punish-neutral}$ = 2.36), indicating a difference in punishment decisions based on the valence of the image. The large standard deviation for the negative images reflects a wide variability in participants' responses to these images, ranging from no punishment to the permanent banning of other social media posters. There were no notable differences with respect to mean punishment amount in the Responsibility ($M_{punish-responsibility}$ = 4.08 days; $SD_{punish-responsibility}$ = 9.01) and Control ($M_{punish-control}$ = 4.13 days; $SD_{punish-control}$ = 8.96) groups, suggesting that the presence of the content moderator responsibility did not substantially influence participants' punishment decisions.

In this regard, a linear mixed regression analysis revealed that both Images (*negative-positive*: OR = 30.89; 95% CI [29.24,32.62]; $p <$ .001; *neutral-positive*: OR = 1.10; 95% CI [1.03,1.19]; $p$ = 0.006) and Poster Intention (*approval-disapproval*: OR = 1.24; 95% CI [1.21,1.26]; $p <$ .001) significantly predicted punishment decisions, while Group did not (OR = 1.03; 95% CI [0.83,1.29]; $p$ = 0.78) (See Table 2 for fixed effects estimates, standard errors, and z-scores). Fig 3A and 3B provides a visual representation of the distribution of punishment across Images and Poster Intention, between Groups. This assignment of punishment indicated that participants were more likely to punish those that had shared content that was morally negative than morally positive or neutral. Participants also assigned higher punishment when the content was shared with the approval of the content poster rather than with disapproval.

**Table 2. Linear mixed regression model output for the likelihood of reporting social media content, showing fixed effects estimates, standard errors, z-values, and p-values.**

| Independent variable | β | SE | z-value | p-value |
|---|---|---|---|---|
| **Image: neutral** | 0.104 | 0.038 | 2.735 | 0.006 ** |
| **Image: negative** | 3.430 | 0.028 | 122.723 | < .001 *** |
| **Poster Intention: approve** | -0.215 | 0.010 | -22.349 | < .001 *** |
| **Group: Control** | -0.031 | 0.113 | -0.277 | 0.78 |

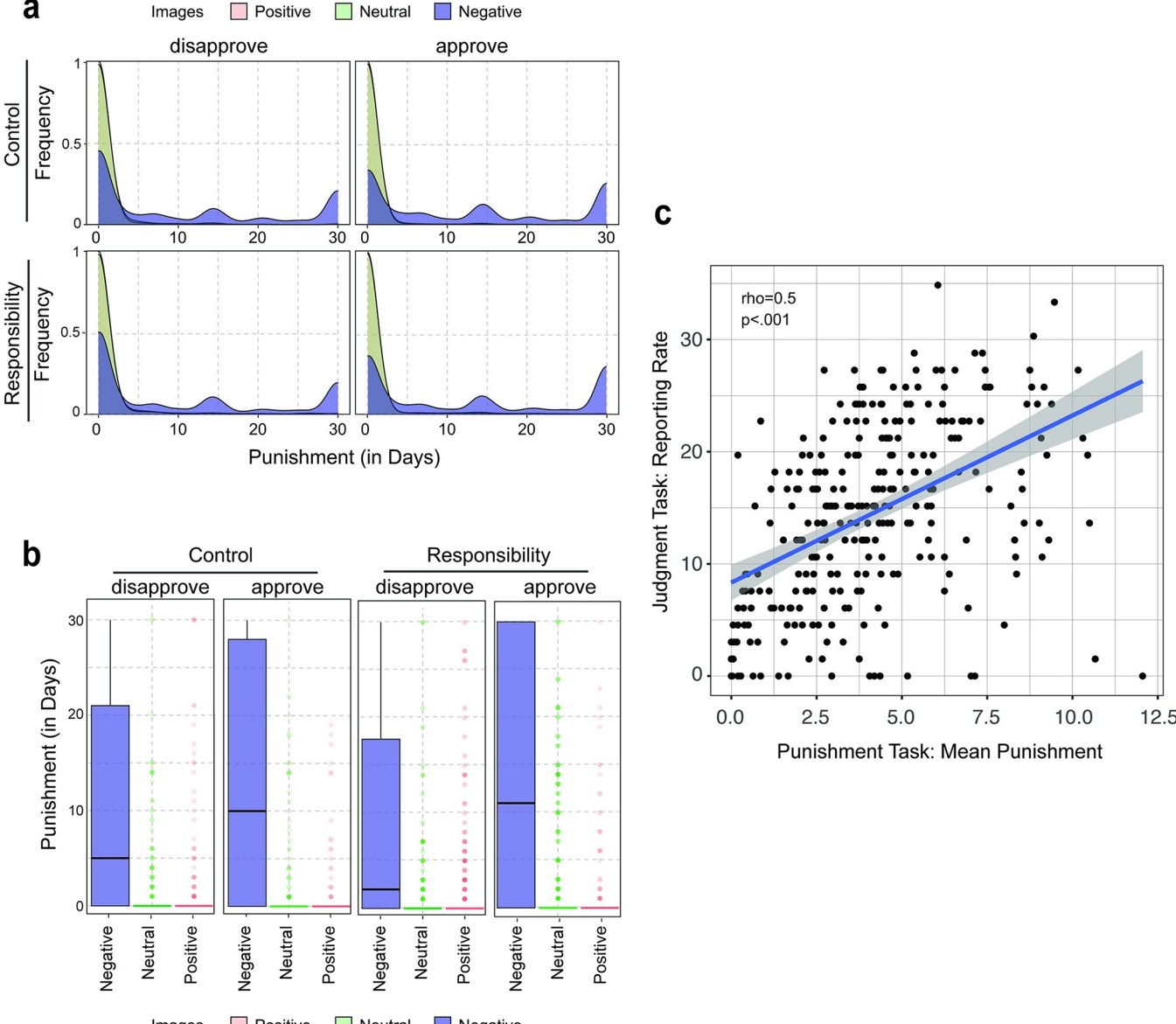

**Fig 3. Punishment task behaviour.** (a) Distribution of punishment (No ban (0 days)–Permanent ban (30+ days)) between Groups, and across Images and Poster Intention. (b) Boxplot depicting punishment score per Image Category (separated into panels according to Poster Intention and Group). The lower and upper hinges correspond to the first and third quartiles (the 25th and 75th percentiles). Jittered data reflects outliers. (c) Scatter plot depicting the relationship between reporting rates (in percentage) in the Judgment Task and mean punishment (in days) in the Punishment Task.

## Correlation between task behaviour

A Spearman's correlation analysis was performed to investigate the relationship between behaviour on the Judgment Task and the Punishment Task. The results showed a strong and statistically significant positive correlation (rho = 0.5, $p < .001$, n = 294) between the two tasks, suggesting that participants who reported more content in the Judgment task, tended to also punish more content posters in the Punishment Task. This relationship is illustrated in Fig 3C.

## Social media related opinions

**Opinions on harmful social media content.** In response to the question: *"I've seen content I would consider inappropriate or harmful on social media"*, 99% of participants answered in the affirmative (9% '*Very Rarely*', 19% '*Rarely*', 50% '*Occasionally*', 14% '*Very Frequently*' and 7% '*Always*') with 1% of participants answering '*Never*'. Furthermore, in response to the question: *"I think those that post content I would consider inappropriate or harmful on social media should be punished"*, 75% of participants answered positively (52% '*Agree*' and 23% '*Strongly Agree*'), 18% were '*Undecided*', and 7% answered negatively (6% '*Disagree*' and 1% '*Strongly*').

**Trust in social media platforms.** Overall, participants expressed low and undecided levels of trust across all social media platforms, with the lowest reported trust observed for Facebook (trust = 15%, distrust = 69%), Tiktok (trust = 14%, distrust = 59%), Instagram (trust = 22%, distrust = 58%), and Twitter/X (trust = 29%, distrust = 42%), as of June 2022. Additionally, frequency of social media use per platform was captured in this study with YouTube (99.7% use; 0.3% do not use), Instagram (90.5% use; 9.5% do not use), Twitter/X (80.3% use; 19.7% do not use), and Facebook (79.6% use; 20.4% do not use) being the most frequently used platforms. Fig 4A shows distribution of participants' trust across 12 popular social media platforms, and Fig 4B displaying the distribution of usage frequency across all 12 platforms.

To investigate the relationship between trust and reporting rates, we examined the percentage of content reported in the Judgment task by participants who expressed trust versus those who expressed distrust in the four most frequently used platforms. Mann-Whitney U tests showed that participants who distrust YouTube, Instagram, Twitter/X, and Facebook had significantly lower reporting rates (i.e. reported fewer images in the Judgment task) than those who trust these platforms. Specifically, for YouTube, participants who trust the platform had a median reporting rate of 15.13%, while those who distrust it had a median reporting rate of 13.64% (U = 5550.5, $p = .007$). Participants who trust Instagram had a median reporting rate of 16.67%, while those who distrust it had a median reporting rate of 13.64% (U = 4523, $p = .027$). For Twitter/X, participants who trust the platform had a median reporting rate of 16.67%, while those who distrust it had a median reporting rate of 13.64% (U = 4015.5, $p = .002$). Finally, participants who trust Facebook had a median reporting rate of 18.18%, while those who distrust it had a median reporting rate of 13.64% (U = 3045, $p < .001$). These results thus suggest that participants who express distrust towards social networking platforms are less likely to report content compared to those who express trust. Fig 4C shows reporting rates in the Judgment task as a function of trust.

**Individual traits.** To explore the relationship between empathy subscales and reporting rates in the Judgment Task, Spearman's rho correlations were conducted. The results revealed a statistically significant positive correlation between empathetic concern (EC) and reporting (rho = 0.17, $p = 0.003$, n = 294) and between perspective taking (PT) and reporting (rho = 0.12, $p = 0.037$, n = 294). On the other hand, the correlation between personal distress (PD) and reporting was non-significant (rho = 0.1, $p = 0.086$, n = 294).

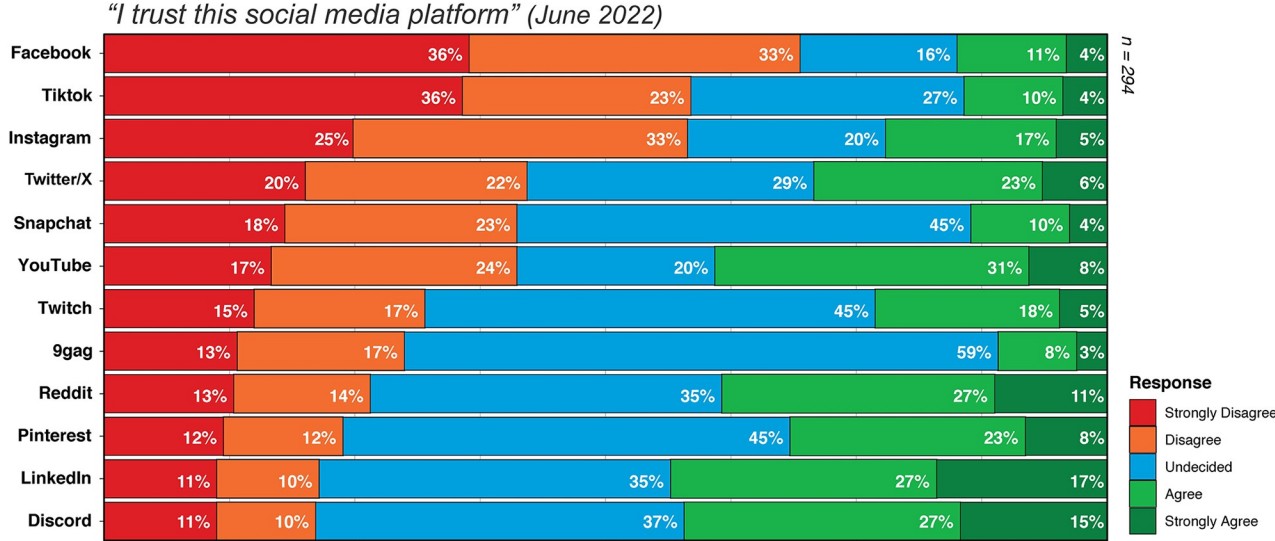

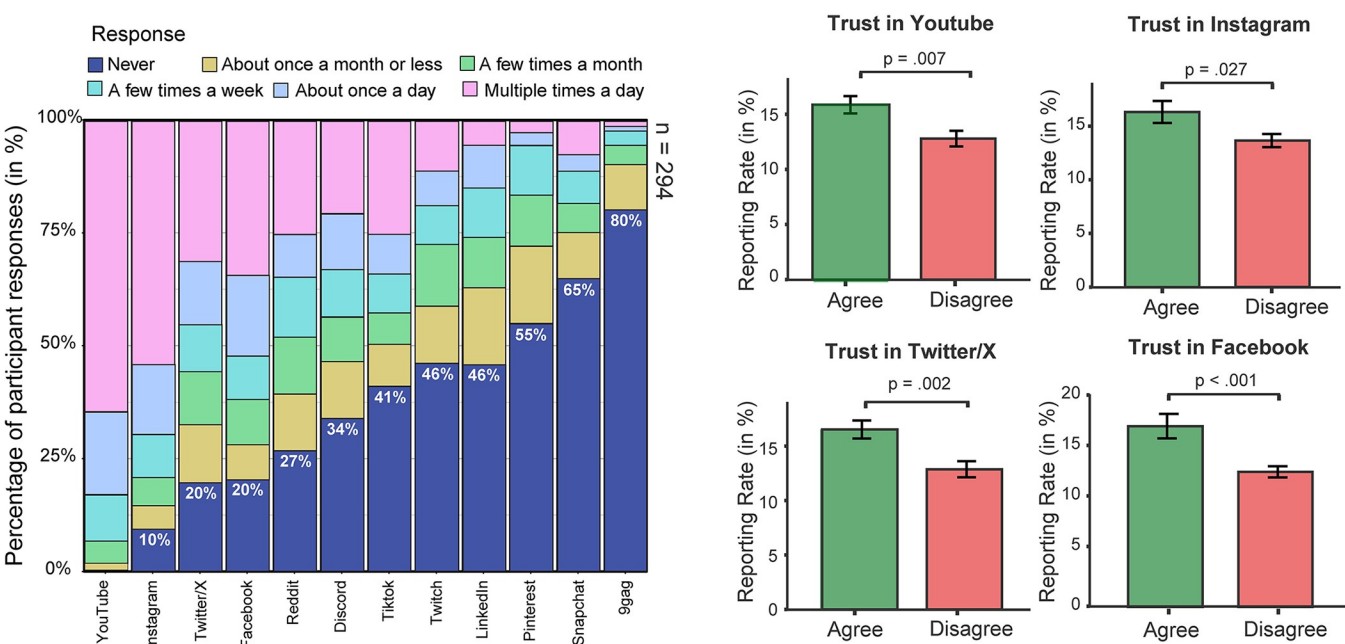

**Fig 4. Trust and use of social media platforms.** (a) Trust in platform (in percentage), captured in response to the question: "I trust this social media platform and believe that the people/company that run and manage it are honest."; (b) Distribution of frequency of use (in percentage) across 12 social media platforms; and (c) Total reporting rate (percentage reported in the Judgment task) as a function of participants trust in YouTube, Instagram, Twitter/X and Facebook.

Similarly, to examine the relationship between empathy subscales and mean punishment in the Punishment Task, Spearman's rho correlations were conducted. The results showed weak, non-significant positive correlations between EC and punishment (rho = 0.11, $p$ = 0.072, n = 294), PT and punishment (rho = 0.1, $p$ = 0.08, n = 294), and PD and punishment

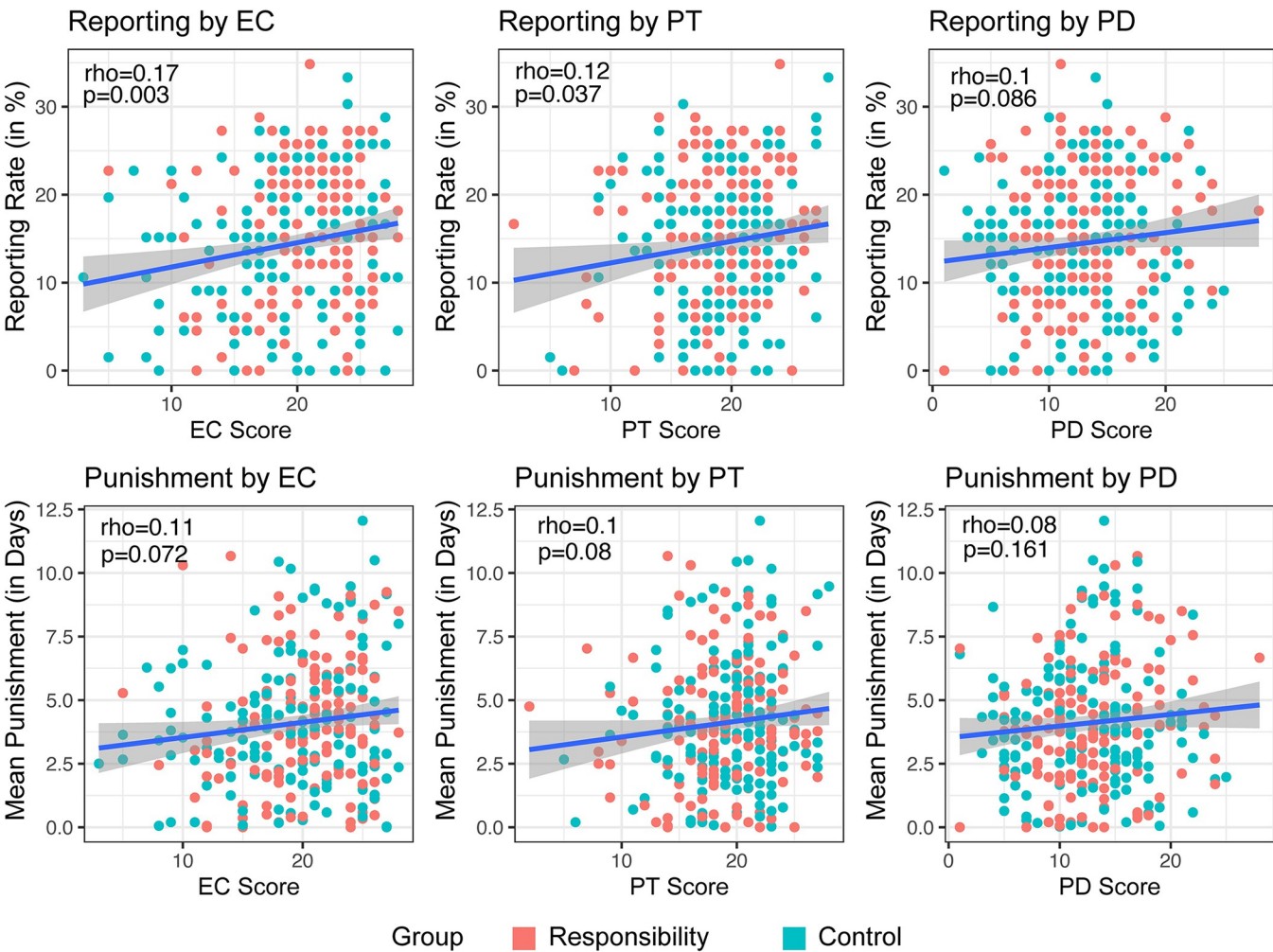

**Fig 5.** Scatter plots depicting relationship between empathy subscales and reporting rates in the Judgment Task (top row) and mean punishment in the Punishment Task (bottom row). Spearman's rho correlations were conducted to explore the association between empathy subscales (Empathetic Concern (EC), Perspective Taking (PT), and Personal Distress (PD)) and reporting rates in the Judgment Task and mean punishment in the Punishment Task respectively.

(rho = 0.082, $p$ = 0.161, n = 294). Fig 5 depicts the relationships between empathy and Judgment and Punishment Tasks respectively. See Supporting Information for further exploratory analyses conducted on individual characteristics.

## Discussion

Social media shapes the interactions and opinions of hundreds of millions of people every day. However, to date we have relatively little understanding as to how individuals determine what is right or wrong in online environments and, in particular, what they decide to do about content they deem objectionable. This question is particularly relevant in light of the prevalence of harm online–in fact, 99% of our study participants reported previously having been exposed to what they deemed to be inappropriate social media content. Accordingly, this study sought to examine factors that influence both individual decisions to report social media content as well as preferences for punishing users who choose to share inappropriate content, tested across a diverse sample of users. To do this, we designed a novel online study and asked participants to

view images previously rated as either morally positive, negative, or neutral. Participants then had to indicate whether they would support removing such content and banning the content posters. Additionally, we investigated whether knowledge regarding the intentions of the poster as well as changing the perceived responsibility of the participant would impact their decisions. Examining how these factors impact users is crucial to better understanding public preferences for content moderation [35], and specifically the role that users play in moderating content on social media.

In both reporting and punishment decisions, we found moral image category to be a driver of objectionability to online content. Specifically, participants almost exclusively chose to report morally negative images as compared to morally positive or neutral images. Negative imagery in our study varied broadly in content and included depictions of violence, riots, racist propaganda and harm being committed to others, animals and the environment. These results thus build on existing literature which suggests that acts of wrongness violate moral norms [14], extending these findings to an online context. Relatedly, participants in our study considered the sharing of morally negative images as an act worthy of punishment, and were willing to ban other users ranging across multiple days, and even permanently. Taken together, our participants clearly demonstrated that sharing depictions of immorality and acts of harm are considered inappropriate on social media. This is further reinforced by self-reported responses in which 75% of our participants indicated that sharing harmful content should indeed be prohibited and punished. These results thus highlight the importance of the moral valence of social media imagery in users' assessments of what is appropriate online. While previous studies have shown the relevance of moral content in the spread and virality of social media content [11], our study highlights the importance of negative moral valence in what images social media users wish to flag and think should be punished.

We also found an effect of poster *intention* in driving both reporting and punishment decisions respectively. Participants flagged more morally negative images which had been posted with the approval of an ostensible content poster than similar images which had been posted with disapproval. Similarly, in determining whether to ban other social media users, participants assigned higher punishments to those who endorsed morally negative images as compared to those who denounced similar content. Accordingly, and in line with previous findings in moral psychology [13, 14, 27], our results suggest that individuals weigh the potential harm caused, with an agent's *intent* to cause such harm in their moral judgments and decisions. Our study sheds light on this in a realistic online setting indicating that in determinations of harm online, it is not only harm *depicted* but harm *intended* that individual users believe to be a relevant feature in content moderation decisions. Our findings are also broadly consistent with literature documenting the 'harm-magnification effect', which describes how perceived intent can motivate individuals to magnify observed harms [16, 36]. This tendency to overestimate the impact of intentional harm is of interest when considering user perceptions about the damage caused by inappropriate content, and merits further research in order to comprehensively understand how users assess intention in online contexts. Accordingly, we show that intention is an important aspect that should therefore be considered in efforts to prioritise and mitigate harmful online content. This is especially of note since platform governance literature shows a shift from the binary paradigm of content moderation (e.g. take-down/keep up) to a more nuanced menu of options supposed to sanction posters and limit the reach of harmful content [37]. As machine learning accuracy for content moderation tasks improves, additional longitudinal features may be taken into account when trying to infer intention, such as an account's actionability track-record.

In respect of participants' *perceived responsibility*, we found an effect in reporting decisions. Specifically, participants who were assigned a 'user content moderation' role exhibited 2.3%

higher reporting rates than participants who had not been assigned this responsibility. Though small, this increase was statistically significant and of course when scaled to actual moderation decisions is of considerable interest. To explain this increase, we draw on bystander intervention and reporting literature which have shown context-dependent altruistic behaviour, such that the intention to report wrongdoing is positively associated with perceived responsibility [19, 20]. However, while reporting harm has been studied in different contexts (for example, individuals' reporting crime to the police or whistleblowing in an organisational context) [20], by focusing on the perceptions of online users, our results advance this literature by suggesting user responsibility as a key psychological mechanism that impacts flagging behaviour in an online setting. That is, while a core assumption underlying the "report" function of existing SNSs is that users will indeed make use of them every time they see problematic content, our study specifically draws attention to the importance of what users *perceive* to be their personal role in their willingness to use this important feature. Our results thus suggest that altering the perceptions of social media users as to their role in the content moderation process can affect their reporting behaviour with more content flagged by those who think they have a greater responsibility to address harm. This effect may potentially be even stronger in the context of individuals who identify as members of the communities they supposedly serve through content moderation, although this remains to be further studied [23].

Interestingly, however, while perceived responsibility had a significant effect on reporting behaviour, it was not similarly predictive of willingness to punish. Participants across both control and responsibility groups indicated similar preferences on what should be punished and the length of ban that they believe posters should receive. Our study thus contradicts some previous, albeit limited, research which has suggested a decrease in altruistic punishment due to diffusion of responsibility [38]. It also adds credence to the theoretical idea that reporting and punishing harm may be distinct psychological processes [14], and thus are important to study independently.

Finally, in order to gain further insight into participants' responses, we explored individual characteristics that may drive responses online. Notably, we observed a significant negative association between trust in social media platforms and reporting decisions. Our sample indicated low levels of trust across all 12 social media platforms we questioned them about. The most popular SNSs were, at time of data collection in June 2022, YouTube, Instagram, Twitter/ X and Facebook. We found that participants who expressed lower levels of trust in each of the four most popular platforms were less likely to report content overall. This relationship is in line with research that suggests an association between trust in authority and likelihood of reporting harm to that authority [25, 26]. In fact, several studies across different contexts have found that individuals who trust the authority responsible for handling reports of harm (for example, police, human resources, etc.) are more likely to report harm when it occurs [26, 39]. We believe that these findings thus have interesting implications for SNSs and regulatory bodies who wish to understand, and ideally increase, user cooperation and willingness to report content on social media platforms. Nonetheless, it is important to note that research on the relationship between trust and reporting harm to authorities is inconsistent [40], and has never been studied in the context of reporting negative content on social media. Therefore, further research is needed to delve deeper into these results and to examine the role of trust in shaping an individual's willingness to flag harmful content on social media, especially in an increasingly complex social media context dominated by authenticity and community-based niches [41, 42].

Similarly, we also examined the relationship between subscales of trait empathy and the likelihood of reporting objectionable content and punishing other users. We found that both empathetic concern, reflecting the tendency to feel sympathy and concern for others, as well as perspective taking, which involves spontaneously adopting the psychological point of view of

others, were both positively associated with reporting rates. However, personal distress, measuring self-orientated feelings in interpersonal settings, did not show a significant association with reporting behaviour. This absence could potentially be attributed to participants not encountering any personal or explicit images during the study, which could have elicited self-orientated feelings of anxiety or distress. Our results thus suggest that users' empathy, particularly pertaining to concern for others, may be an important factor when it comes to their reporting of objectionable content on social media. The impact of empathy on preferences to administer punishment were more limited which, once again, suggest that distinct psychological processes underlie reporting and punitive choices. While our findings are generally in line with prior research [24], further work is needed to comprehensively explore the role of individual differences in shaping views of objectionability on social media.

Further limitations on the generalisability of the findings should be noted. One inherent in our experimental design is the awareness among participants that their decisions to report posts or enact punishment were hypothetical. The absence of real-world consequences for participants' choices confines the ecological validity of our findings and may have impacted the observed outcomes, potentially contributing to the relatively subtle differences in reporting rates between responsibility and control groups. In a similar vein, additional research would benefit from investigating actual social media content and other realistic contextual factors that may shape user experiences and be key drivers of individual responses. Furthermore, while our investigation into perceived responsibility opens the door to further research examining the bystander framework in online environments, future studies are needed to explore the various factors and incentives which may impact perceptions of objectionability. Such research would play an essential role in comprehensively understanding individual motivations online and in devising strategies that platforms and regulatory bodies could employ to empower users to exercise their rights.

As one of the first empirical studies to investigate user' decisions to report and punish others on social media, our findings extend existing research on moral judgment by elucidating factors that impact individual assessments of objectionability online. Our study underscores the importance of moral valence in users' views of social media content moderation, and highlights the importance of considering the intention behind content creation in efforts to prioritize and mitigate perceived online harm. In addition, this study advances understanding of prosocial behaviour online by investigating the role of perceived responsibility on bystander intervention on social media. Our findings indicate that altering the perceptions of social media users regarding their role in the content moderation process has a tangible impact on their reporting behaviour. We also raise critical questions regarding individual differences, including the importance of trust in social media platforms and empathy in order to understand users' willingness to engage in mechanisms of content moderation. Overall, our study provides social media platforms and those involved in the development of content moderation policies with empirical evidence to better understand the nuances of social media users' decision-making in response to various online content. Ultimately, whilst it is important to understand the role and responsibilities of each social media platform in moderating harmful content, it is equally crucial to gain insights into users' perceptions and preferences in the pursuit of creating safer online environments.

## Supporting information

**S1 File.** This file contains various supporting materials, methods and analyses, including, (1) Moral Images: This section details the images used in the Judgment and Punishment Tasks respectively, including the 'Moral Wrongfulness' ratings for each individual image (S1 and S2 Tables); (2) Questionnaires: This section presents each question that participants completed;

and (3) Exploratory Analyses: This section includes exploratory graphs on participant responses as function of political orientation (S3 Table and S2 Fig), and current country of residence (S4 Table, S3 and S4 Figs).
(DOCX)

## Acknowledgments

The authors are grateful to the members of the Decision Neuroscience Lab (Donders Institute for Brain, Cognition, and Behaviour, the Radboud University Nijmegen), as well as Stefan Prekovic for their helpful comments on this study.

## Author Contributions

**Conceptualization:** Sarah Vahed, Catalina Goanta, Pietro Ortolani, Alan G. Sanfey.

**Data curation:** Sarah Vahed.

**Formal analysis:** Sarah Vahed.

**Funding acquisition:** Catalina Goanta, Alan G. Sanfey.

**Investigation:** Sarah Vahed.

**Methodology:** Sarah Vahed.

**Supervision:** Catalina Goanta, Pietro Ortolani, Alan G. Sanfey.

**Visualization:** Sarah Vahed.

**Writing – original draft:** Sarah Vahed.

**Writing – review & editing:** Sarah Vahed, Catalina Goanta, Pietro Ortolani, Alan G. Sanfey.

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
