## [Decision Letter · Decision Letter 0]

13 Dec 2023

PONE-D-23-28475

Moral judgment of objectionable online content: Investigating factors influencing reporting decisions and punishment preferences on social media

PLOS ONE

Dear Dr. Vahed,

Thank you for submitting your manuscript to PLOS ONE. After careful consideration, we feel that it has merit but does not fully meet PLOS ONE’s publication criteria as it currently stands. Therefore, we invite you to submit a revised version of the manuscript that addresses the points raised during the review process.

We look forward to receiving your revised manuscript.

Kind regards,

Yasuko Kawahata

Academic Editor

PLOS ONE

4. We note that Figure 1 in your submission contain copyrighted images. All PLOS content is published under the Creative Commons Attribution License (CC BY 4.0), which means that the manuscript, images, and Supporting Information files will be freely available online, and any third party is permitted to access, download, copy, distribute, and use these materials in any way, even commercially, with proper attribution. For more information, see our copyright guidelines: http://journals.plos.org/plosone/s/licenses-and-copyright.

5. We note that Figure S1 in your submission contain [map/satellite] images which may be copyrighted. All PLOS content is published under the Creative Commons Attribution License (CC BY 4.0), which means that the manuscript, images, and Supporting Information files will be freely available online, and any third party is permitted to access, download, copy, distribute, and use these materials in any way, even commercially, with proper attribution. For these reasons, we cannot publish previously copyrighted maps or satellite images created using proprietary data, such as Google software (Google Maps, Street View, and Earth). For more information, see our copyright guidelines: http://journals.plos.org/plosone/s/licenses-and-copyright.

a. You may seek permission from the original copyright holder of Figure S1 to publish the content specifically under the CC BY 4.0 license.  

Reviewers' comments:

Reviewer's Responses to Questions

Comments to the Author

1. Is the manuscript technically sound, and do the data support the conclusions?

Reviewer #1: Yes

Reviewer #2: Yes

2. Has the statistical analysis been performed appropriately and rigorously?

Reviewer #1: Yes

Reviewer #2: Yes

3. Have the authors made all data underlying the findings in their manuscript fully available?

Reviewer #1: Yes

Reviewer #2: Yes

4. Is the manuscript presented in an intelligible fashion and written in standard English?

Reviewer #1: Yes

Reviewer #2: Yes

5. Review Comments to the Author

Reviewer #1: This study aimed to elucidate factors that drive individuals’ assessment of objectionability to online content in a diverse sample of users (n=294). Specifically, the authors examined the impact of another users’ intention behind sharing an image on of perceived and the responsibility on user decisions to report content to platforms and preferences to assign punishment to other social media users. The findings reveal that moral imagery strongly influences users' assessments of what is appropriate online content, with participants almost exclusively choosing to report and punish morally negative images. Poster intention also plays a significant role in user’s decisions to report content and assign punishment, with participants objecting more to negative content which had been supported by another content poster. Increased perceived responsibility affected reporting behaviour but not punishment preferences.

Abstract

Please further explain the aim and research questions of the study.

Introduction

Please further clarify hypotheses presented. While reading your work, I believe a more in-depth explanation and stronger ties to the existing literature would enhance the overall robustness of your research. It would be immensely beneficial if you could provide a more detailed rationale for each hypothesis, explicitly connecting them to relevant studies cited in your work.

Methods

Authors stated that they recruited “a diverse sample”. I support the inclusion of a diverse sample in this type of studies. However, I consider that further explanation regarding factors leading to diversity the sample are required. I addition, considering the sample diversity, I could be interesting to explore whether demographic characteristics or political preferences are associated with any of the dependent variables.

Why did the authors report the participants’ demographic characteristics in Supplementary Material? I suggest including them into the Participants section.

Please report the respective power calculation and the number of participants assigned to each group.

Please explain the reason for excluding the “Fantasy” scale of the IRI.

Discussion

Results of this study are also in line with the called “harm-magnification effect”. It could be interesting to discuss the current results in the light of the previous literature on this effect. Please see:

https://doi.org/10.1073/pnas.1501592112

https://doi.org/10.1057/s41599-020-00595-8

Authors should further explain and discuss the findings on the relationship between empathy and the likelihood of reporting objectionable content and punishing other users. Please also discuss the lack of association between personal distress and users’ behavioral results.

Implications and relevance of the results of this study should be further discussed.

Reviewer #2: The sumitted work examined factors that drive individuals to report objectionable online content and punish posters of such content in SNSs. In the experiment, the participants were presented images of negative, neutral and positive moral valence in the form of a social media post. The post was accompanied by approving or disapproving expression from the supposed content poster. The participants were randomly assigned to a group of a regular user or "user content moderator". The influence of these three factors (Image, Poster Intention, and Responsibility) were measured on decisions to report content to platforms

and preferences to assign punishment to the content posters. Unsuprisingly, the participants almost exclusively chose to report morally negative images and punish the posters of such content. Poster approval of the content increased the users' decisions to report content and assign punishment. The Responsibility or role in which the participant went through the content affected reporting behaviour but not punishing. Moreover, a negative association between trust in SNSs and reporting decisions was observed, as well as a positive relationship between trait empathy and reporting rates.

The article is well-written and the argumentation and justification of the paper is solid. The statistical analyses seem solid although some of the methological choises could be justified in the method section. The only more major thing needing acknowledgement is the limitation related to the experimental task presented to the participants. That is, since the participants were aware that they were not actually reporting any posts of punishing the posters and since the participant's role (content moderator, regular user) were hypothetical, the study lacks ecological validity and some of the results (e.g., the very small difference in reporting between Responsible and Contro group) could be explained by the fact that the decisions to report were considered as hypothetical and not having real consecuences on real posters. This is, of course, a limitation that could be acknowledged in the discussion section. There is a passage on page 20 that discusses generalisability of the findings and limitations in that regard by the authors could be more explicit about the implied limitation.

Minor points:

Page 3 (first paragraph of intro): The authors speak about the consecuences of being exposed to objectionable online content in social media but the causal relation from the exposure to, for instance, to dissemination of fake news, elevation of extreme violence etc. is likely not one-directional. That is, there can be a correlation between the things but there might not necessarily be the supposed causal link. Therefore, I would recommend taking this into account in the argumentation.

Page 11 (statistical analyses): The authors report that in the model predicting punishment intensity, a glmm with gamma distribution was used. Please clarify why was the DV assumed to follow gamma distribution rather than gaussian distribution. Also, why was a multilevel logistic regression with binominal link function used when predicting the outcome of reporting of the the response had three values: like, dislike, and report. Although there is probably a good reason for this, please clarify the decision in the text.

Page 15: The authors use non-parametric texts in many contexts. Were the assumptions of parametric tests always violated or why was non-parametric option adopted so often?

6. PLOS authors have the option to publish the peer review history of their article (what does this mean?). If published, this will include your full peer review and any attached files.

Do you want your identity to be public for this peer review? For information about this choice, including consent withdrawal, please see our Privacy Policy.

Reviewer #1: No

Reviewer #2: No

---

## [Author Response · Author response to Decision Letter 0]

28 Dec 2023

Rebuttal Letter 

Reviewer #1: This study aimed to elucidate factors that drive individuals’ assessment of objectionability to online content in a diverse sample of users (n=294). Specifically, the authors examined the impact of another users’ intention behind sharing an image on of perceived and the responsibility on user decisions to report content to platforms and preferences to assign punishment to other social media users. The findings reveal that moral imagery strongly influences users' assessments of what is appropriate online content, with participants almost exclusively choosing to report and punish morally negative images. Poster intention also plays a significant role in user’s decisions to report content and assign punishment, with participants objecting more to negative content which had been supported by another content poster. Increased perceived responsibility affected reporting behaviour but not punishment preferences.

Abstract

Please further explain the aim and research questions of the study.

We appreciate the reviewer's feedback and have now made appropriate revisions to the abstract to ensure our study’s aims and research questions are explicit.

Introduction 

Please further clarify hypotheses presented. While reading your work, I believe a more in-depth explanation and stronger ties to the existing literature would enhance the overall robustness of your research. It would be immensely beneficial if you could provide a more detailed rationale for each hypothesis, explicitly connecting them to relevant studies cited in your work.

We are grateful for the reviewer's insightful comments and recognise the importance of more detailed explanations of our hypotheses, along with stronger ties to the existing literature. We have now made several refinements to the Introduction which aim to strengthen the connection between our hypotheses and relevant studies cited (pages 4 – 8). In addition, we have specifically introduced a paragraph which sets out our three hypotheses with explicit reference to the related literature (page 9). For ease of reference, this paragraph is copied below: 

[Page 9]: 

“Given the growing need to address online harm, and the limited empirical work that directly investigates factors driving social media users’ objectionability to content, this study specifically seeks to answer the following research questions: What type of moralised content do individuals think should be reported and punished on social media? What is the role of intention behind sharing an image on reporting and punishment decisions? Does an individual’s sense of responsibility impact decisions to flag content and punish other users? 

Based on prior work which has highlighted the prioritisation of morally polarising content (12, 30), our hypothesis is that users focus on moral messaging, particularly those which are morally negative, when making decisions to object to online content. Moreover, given the abundance of research which has emphasised the importance of perceived intent in moral judgments (16, 29), we hypothesise that poster intention will emerge as a significant determinant in assessments of online objectionability. Finally, acknowledging the robust impact of the bystander effect (20, 21), we anticipate that altering the mindset of the user as to their role in content governance will impact their willingness to intervene through moderation mechanisms. We additionally investigate how these user decisions and preferences may be understood by individual characteristics of social media users, such as their trust in SNSs and their empathy.” 

Methods

Authors stated that they recruited “a diverse sample”. I support the inclusion of a diverse sample in this type of studies. However, I consider that further explanation regarding factors leading to diversity the sample are required. I addition, considering the sample diversity, it could be interesting to explore whether demographic characteristics or political preferences are associated with any of the dependent variables.

Why did the authors report the participants’ demographic characteristics in Supplementary Material? I suggest including them into the Participants section.

We appreciate the reviewer's attention to, and support of, the diverse sample recruited for the study, and have now further clarified and explained our choice for a diverse European sample in the Methods section (page 10): 

Page 10: “This criteria ensured we obtained a representative sample of social media users across diverse countries in the EU that are subject to content governance legislation, in this case, the DSA.”

We further are grateful for the suggestion of exploring relevant demographics or political preferences. Our Supporting Information (referred to on page 21) now includes these exploratory analyses. We specifically wish to draw the reviewer’s attention to the data on political orientation, which demonstrates the homogeneity of political preference across the study sample. This uniformity unfortunately precludes us from conducting analyses aimed at assessing variations in dependent variables across political lines. 

Finally, we now include details on participant’s demographic characteristics directly in the Methods section (page 10):

Page 10: “All of the remaining participants (n=294; 170 males, 123 females and 1 ‘prefer not to say’; Mage=28.33 years, SDage=8.42) were included in the analyses (sample size per ‘Group’ as defined below: Control: n=149; Responsibility: n=145).”

Please report the respective power calculation and the number of participants assigned to each group.

We conducted a power analysis using G*Power 3.1 to determine the optimal sample size for robustly detecting meaningful effects within our study. The results of this analysis revealed that a total sample size of 119 participants would be necessary to achieve a substantial statistical power of 0.95. In our calculations, we assumed an effect size of 0.15, a significance level of 0.05, and a model featuring three test predictors. The analysis was conducted within the framework of the family of F-tests. Considering the absence of prior research on reporting and punishment decisions on social media, we adopted a conservative approach by collecting a sample size of 150 participants per between-subject group. We obtained an equitable distribution of participants per group, with 149 participants (1 excluded) in the Control group and 145 participants (5 excluded) in the Responsibility group.

We have now ensured that the above information is clearly reported in the Methods section of the manuscript (pages 9 and 10):

Pages 9 and 10: “The sample size was calculated using G*Power 3.1 (31), which estimated n=119 to achieve a power of 0.95, with significance level of p=0.05, and a model featuring three test predictors. Considering the absence of prior research on reporting and punishment decisions on social media, we opted for a conservative approach by collecting a sample size of 150 per between-subject group. 300 participants were recruited through Prolific (www.prolific.co), and directed to an online experiment programmed on Gorilla (https://gorilla.sc).”

Page 10: “All of the remaining participants (n=294; 170 males, 123 females and 1 ‘prefer not to say’; Mage=28.33 years, SDage=8.42) were included in the analyses (sample size per ‘Group’ as defined below: Control: n=149; Responsibility: n=145).”

Please explain the reason for excluding the “Fantasy” scale of the IRI.

We thank the reviewer for this question, and the opportunity to clarify. The decision to exclude the 'Fantasy' scale from the Interpersonal Reactivity Index (IRI) was made after careful consideration, and is in line with prior research which has similarly excluded this subscale when studying empathy in this context (see: Kardos, P., Leidner, B., Pléh, C., Soltész, P., & Unoka, Z. (2017). Empathic people have more friends: Empathic abilities predict social network size and position in social network predicts empathic efforts. Social Networks, 50, 1-5.). 

The 'Fantasy' scale measures the tendency to place oneself imaginatively into the feelings and actions of fictitious characters. While this scale is relevant to empathy, we chose to focus on other components of empathy that more directly align with our objective of capturing responses to morally diverse online content. Accordingly, we decided to streamline the assessment and interpretation of empathy in the context of our specific research questions. We have now made the rationale behind this choice more clear in the Methods section (on page 13):

Page 13: “IRI’s fourth subscale of ‘Fantasy’, which assesses the ability to imagine and experience the emotions of fictitious characters, was not relevant to our topic of interest and was not included.”

Discussion

Results of this study are also in line with the called “harm-magnification effect”. It could be interesting to discuss the current results in the light of the previous literature on this effect. Please see:

https://doi.org/10.1073/pnas.1501592112

https://doi.org/10.1057/s41599-020-00595-8

We are grateful for the reviewer's guidance to discuss our study's results, specifically our significant findings regarding poster intention, in the context of the 'harm-magnification effect'. The recommended papers highlight how perceived intent motivates individuals to magnify observed harm, resulting in a tendency to overestimate the impact of such (intentional) harm. This observation is particularly interesting when thinking about user responses to intentionally posted negative content, and their perceptions about the damage such content could cause. These recommendations thus assists us in further contextualising our findings on the role of intention on objectionability online, and highlight how our study contributes to the broader literature on intention assessment. We have now referenced these papers in the article as well as included the following addition to our Discussion (page 23):

Page 23: “Our findings are also broadly consistent with literature documenting the ‘harm-magnification effect’, which describes how perceived intent can motivate individuals to magnify observed harms (17, 37). This tendency to overestimate the impact of intentional harm is of interest when considering user perceptions about the damage caused by inappropriate content, and merits further research in order to comprehensively understand how users assess intention in online contexts.”

Authors should further explain and discuss the findings on the relationship between empathy and the likelihood of reporting objectionable content and punishing other users. Please also discuss the lack of association between personal distress and users’ behavioral results.

We agree with this point, and have revised the Discussion section which details these exploratory findings (page 26). We now discuss and provide possible explanations regarding these findings.

Page 26: “Similarly, we also examined the relationship between subscales of trait empathy and the likelihood of reporting objectionable content and punishing other users. We found that both empathetic concern, reflecting the tendency to feel sympathy and concern for others, as well as perspective taking, which involves spontaneously adopting the psychological point of view of others, were both positively associated with reporting rates. However, personal distress, measuring self-orientated feelings in interpersonal settings, did not show a significant association with reporting behaviour. This absence could potentially be attributed to participants not encountering any personal or explicit images during the study, which could have elicited self-orientated feelings of anxiety or distress. Our results thus suggest that users' empathy, particularly pertaining to concern for others, may be an important factor when it comes to their reporting of objectionable content on social media. The impact of empathy on preferences to administer punishment were more limited which, once again, suggest that distinct psychological processes underlie reporting and punitive choices. While our findings are generally in line with prior research (26), further work is needed to comprehensively explore the role of individual differences in shaping views of objectionability on social media.”

Implications and relevance of the results of this study should be further discussed.

As a response to this helpful suggestion, we have made several refinements to our Discussion, particularly as it pertains to the implications of our results. We have also expanded the conclusions section of the manuscript (page 27) to provide a more clear explanation of our studies potential relevance in regard to social media platforms and those involved in content moderation policies. 

Page 27: “As one of the first empirical studies to investigate user’ decisions to report and punish others on social media, our findings extend existing research on moral judgment by elucidating factors that impact individual assessments of objectionability online. Our study underscores the importance of moral valence in users’ views of social media content moderation, and highlights the importance of considering the intention behind content creation in efforts to prioritize and mitigate online harm. In addition, this study advances understanding of prosocial behaviour by investigating the role of perceived responsibility on bystander intervention on social media. Our findings indicate that altering the perceptions of social media users regarding their role in the content moderation process has a tangible impact on their reporting behaviour. We also raise critical questions regarding individual differences, including the importance of trust in social media platforms and empathy in order to understand users’ willingness to engage in mechanisms of content moderation. Overall, our study provides social media platforms and those involved in the development of content moderation policies with empirical evidence to better understand the nuances of social media users’ decision-making in response to various online content. Ultimately, whilst it is important to understand the role and responsibilities of each social media platform in moderating harmful content, it is equally crucial to gain insights into users’ perceptions in the pursuit of creating safer online environments.”

Reviewer #2: The submitted work examined factors that drive individuals to report objectionable online content and punish posters of such content in SNSs. In the experiment, the participants were presented images of negative, neutral and positive moral valence in the form of a social media post. The post was accompanied by approving or disapproving expression from the supposed content poster. The participants were randomly assigned to a group of a regular user or "user content moderator". The influence of these three factors (Image, Poster Intention, and Responsibility) were measured on decisions to report content to platforms and preferences to assign punishment to the content posters. Unsuprisingly, the participants almost exclusively chose to report morally negative images and punish the posters of such content. Poster approval of the content increased the users' decisions to report content and assign punishment. The Responsibility or role in which the participant went through the content affected reporting behaviour but not punishing. Moreover, a negative association between trust in SNSs and reporting decisions was observed, as well as a positive relationship between trait empathy and reporting rates.

The article is well-written and the argumentation and justification of the paper is solid. The statistical analyses seem solid although some of the methological choises could be justified in the method section. The only more major thing needing acknowledgement is the limitation related to the experimental task presented to the participants. That is, since the participants were aware that they were not actually reporting any posts of punishing the posters and since the participant's role (content moderator, regular user) were hypothetical, the study lacks ecological validity and some of the results (e.g., the very small difference in reporting between Responsible and Contro group) could be explained by the fact that the decisions to report were considered as hypothetical and not having real consecuences on real posters. This is, of course, a limitation that could be acknowledged in the discussion section. There is a passage on page 20 that discusses generalisability of the findings and limitations in that regard by the authors could be more explicit about the implied limitation.

We thank the reviewer for their insightful feedback. We acknowledge the reviewers valid comments regarding the hypothetical nature of the tasks and the resulting potential limitations of the findings. To address this valuable input, we have now included this specific limitation (pages 26 and 27) by explicitly highlighting the hypothetical decision of the participant in both content moderator or regular user roles, and the potential implications of this on our findings. For ease of reference, this addition is copied below:

Pages 26 and 27: “One [limitation] inherent in our experimental design is the awareness among participants that their decisions to report posts or enact punishment were hypothetical. The absence of real-world consequences for participants’ choices confines the ecological validity of our findings and may have impacted the observed outcomes, potentially contributing to the relatively subtle differences in reporting rates between responsibility and control groups.”

Minor points:

Page 3 (first paragraph of intro): The authors speak about the consequences of being exposed to objectionable online content in social media but the causal relation from the exposure to, for instance, to dissemination of fake news, elevation of extreme violence etc. is likely not one-directional. That is, there can be a correlation between the things but there might not necessarily be the supposed causal link. Therefore, I would recommend taking this into account in the argumentation.

We thank the reviewer for this helpful suggestion. We have now made several amendments to our introduction (page 3) which ensure that we do not claim a causal link between negative social media content and the listed harmful consequences. 

Page 11 (statistical analyses): The authors report that in the model predicting punishment intensity, a glmm with gamma distribution was used. Please clarify why was the DV assumed to follow gamma distribution rather than gaussian distribution. Also, why was a multilevel logistic regression with binominal link function used when predicting the outcome of reporting of the the response had three values: like, dislike, and report. Although there is probably a good reason for this, please clarify the decision in the text.

We appreciate the reviewer's close attention to our statistical analyses, particularly the choice of distribution and models reported now on pages 14 and 15. 

The reviewer correctly notes the use of a Generalized Linear Mixed Model (GLMM) with a gamma distribution in predicting punishment intensity. The decision to employ a gamma distribution was motivated by several reasons. Firstly, since participants could choose punishment durations ranging in days, the gamma distribution, which is continuous and defined for non-negative values, is appropriate for modelling such data. Importantly, our data did not meet the assumptions of a Gaussian distribution due to the fact that participants in our study did not consistently choose to punish. When punishment was selected, it was almost exclusively in response to the morally negative image category and clustered around three time points, one day, 15 days and 30 days, resulting in a skewed distribution. The gamma distribution, which can exhibit skewness, was thus deemed best to capture the shape of our data compared to a normal Gaussian distribution.

The reviewer is also correct in noting the use of a multilevel logistic regression with a binomial link function for predicting the outcome of reporting. We clarify that in this instance the two choice options ‘like’ and ‘dislike’ were combined into one (which we term ‘Not Report’), which allowed us to run the regression on a binary dependent variable (‘Report’ versus ‘Not Report’). Furthermore, the approach of a multilevel regression was chosen to appropriately handle the categorical nature of the dependent variable and account for the hierarchical structure of the data, where observations are nested within participants. 

To clarify these methodological choices, we have elaborated on the above points in the paragraph ‘Statistical Analysis’ (on pages 14 and 15):

Page 14: “In respect of the Judgment Task, we fitted a multilevel logistic regression to analyse the decision to report (binary: Report versus Not Report (i.e. ‘Like’ and ‘Dislike’ choices combined)).”

Page 15: “The [gamma] distribution choice was made due to its ability to capture the skewed nature of the dataset.”

Page 15: The authors use non-parametric texts in many contexts. Were the assumptions of parametric tests always violated or why was non-parametric 

option adopted so often?

We thank the reviewer for this thoughtful inquiry into our use of non-parametric tests. As the reviewer correctly identifies, the assumptions of parametric tests were not met in both the judgment and the punishment tasks. This was primarily due to the fact that participants did not consistently choose to report or punish content. When they did engage in these actions, it was almost exclusively in response to the morally negative image category. To address these points, we have now explicitly highlighted this in our Methods (page 14), acknowledging the deviation from parametric test assumptions. 

Page 14: “In instances where assumptions for parametric tests were violated, non-parametric equivalents were employed.”

---

## [Decision Letter · Decision Letter 1]

24 Jan 2024

PONE-D-23-28475R1Moral judgment of objectionable online content: Reporting decisions and punishment preferences on social mediaPLOS ONE

Dear Dr. Vahed,

Thank you for submitting your manuscript to PLOS ONE. After careful consideration, we feel that it has merit but does not fully meet PLOS ONE’s publication criteria as it currently stands. Therefore, we invite you to submit a revised version of the manuscript that addresses the points raised during the review process.

We look forward to receiving your revised manuscript.

Kind regards,

Yasuko Kawahata

Academic Editor

PLOS ONE

Journal Requirements:

Reviewers' comments:

Reviewer's Responses to Questions

**Comments to the Author**

1. If the authors have adequately addressed your comments raised in a previous round of review and you feel that this manuscript is now acceptable for publication, you may indicate that here to bypass the “Comments to the Author” section, enter your conflict of interest statement in the “Confidential to Editor” section, and submit your "Accept" recommendation.

Reviewer #2: All comments have been addressed

2. Is the manuscript technically sound, and do the data support the conclusions?

Reviewer #2: Yes

3. Has the statistical analysis been performed appropriately and rigorously? 

Reviewer #2: Yes

4. Have the authors made all data underlying the findings in their manuscript fully available?

Reviewer #2: Yes

5. Is the manuscript presented in an intelligible fashion and written in standard English?

Reviewer #2: Yes

6. Review Comments to the Author

Reviewer #2: I have now read the revised manuscript and the authors' responses to the review comments. I think the comments and suggestions have been addressed adequately and I recommend accepting the manuscipt for publication.

7. PLOS authors have the option to publish the peer review history of their article (what does this mean?). If published, this will include your full peer review and any attached files.

Reviewer #2: **Yes: **Ville Harjunen

---

## [Author Response · Author response to Decision Letter 1]

25 Jan 2024

Rebuttal Letter 

Journal Requirements:

We thank the editor for this comment. We have ensured that none of the references cited in our manuscript have been retracted. In addition, we have ensured that each of the citations are edited according to PLoS ONE’s style/journal requirements. To ensure that our list is complete and concise, we have removed the following two references from the manuscript: McLoughlin, K. L., Brady, W. J., & Crockett, M. J. (2021). The role of moral outrage in the spread of misinformation (conference proceeding abstract, previously labelled as reference number 13 on lines 97 and 105,) and Tucker, J. A., Guess, A., Barberá, P., Vaccari, C., Siegel, A., Sanovich, S., ... & Nyhan, B. (2018). Social media, political polarization, and political disinformation: A review of the scientific literature. Political polarization, and political disinformation: a review of the scientific literature (March 19, 2018) (previously labelled as reference number 8 on line 61). 

Reviewers' comments:

Review Comments to the Author

Reviewer #2: I have now read the revised manuscript and the authors' responses to the review comments. I think the comments and suggestions have been addressed adequately and I recommend accepting the manuscript for publication.

We sincerely appreciate the reviewers thorough evaluation of both the original and revised manuscript, as well as our responses to previous comments.

---

## [Decision Letter · Decision Letter 2]

7 Mar 2024

Moral judgment of objectionable online content: Reporting decisions and punishment preferences on social media

PONE-D-23-28475R2

Dear Dr. Sarah Vahed,

We’re pleased to inform you that your manuscript has been judged scientifically suitable for publication and will be formally accepted for publication once it meets all outstanding technical requirements.

Kind regards,

Yasuko Kawahata

Academic Editor

PLOS ONE

Additional Editor Comments (optional):

Reviewers' comments:

Reviewer's Responses to Questions

**Comments to the Author**

1. If the authors have adequately addressed your comments raised in a previous round of review and you feel that this manuscript is now acceptable for publication, you may indicate that here to bypass the “Comments to the Author” section, enter your conflict of interest statement in the “Confidential to Editor” section, and submit your "Accept" recommendation.

Reviewer #2: All comments have been addressed

Reviewer #3: (No Response)

2. Is the manuscript technically sound, and do the data support the conclusions?

Reviewer #2: Yes

Reviewer #3: Yes

3. Has the statistical analysis been performed appropriately and rigorously? 

Reviewer #2: Yes

Reviewer #3: Yes

4. Have the authors made all data underlying the findings in their manuscript fully available?

Reviewer #2: Yes

Reviewer #3: Yes

5. Is the manuscript presented in an intelligible fashion and written in standard English?

Reviewer #2: Yes

Reviewer #3: Yes

6. Review Comments to the Author

Reviewer #2: (No Response)

Reviewer #3: This research investigated people’s moral responses to different types of content on social media. Specifically, it examined the influence of the moral imagery, intention of a content poster, and perceived responsibility of social media users on participants’ reporting and punishment decisions. Individual user characteristics were also taken into account. After two rounds of revisions, the paper is well-written and well-illustrated. I believe it can make a valuable contribution to the knowledge of moral judgment. I only have some minor concerns.

1. It is assumed that the ‘Responsibility’ group has a stronger sense of responsibility than the ‘Control’ group. However, a manipulation check is missing. It should be mentioned in the Discussion section as a limitation.

2. It seems that there is an overlap or occlusion issue in Figure 3a, making it difficult to see the curve for the positive and neutral groups. The author may consider adjusting the transparency or using arrows for annotation to improve visibility.

3. In the punishment task, how many moral images of each type were used? What were the criteria for selecting the images?

7. PLOS authors have the option to publish the peer review history of their article (what does this mean?). If published, this will include your full peer review and any attached files.

Reviewer #2: **Yes: **Dr. Ville J. Harjunen

Reviewer #3: No

---

## [Editor Report · Acceptance letter]

14 Mar 2024

PONE-D-23-28475R2 

PLOS ONE

Dear Dr. Vahed, 

I'm pleased to inform you that your manuscript has been deemed suitable for publication in PLOS ONE. Congratulations! Your manuscript is now being handed over to our production team.

Kind regards, 

on behalf of

Dr. Yasuko Kawahata 

Academic Editor

PLOS ONE